# Surveillance of Viral Encephalitis in the Context of COVID-19: A One-Year Observational Study among Hospitalized Patients in Dakar, Senegal

**DOI:** 10.3390/v14050871

**Published:** 2022-04-22

**Authors:** Jamil Kahwagi, Al Ousseynou Seye, Ahmadou Bamba Mbodji, Rokhaya Diagne, El hadji Mbengue, Maouly Fall, Soa Fy Andriamandimby, Ava Easton, Martin Faye, Gamou Fall, Ndongo Dia, Babacar Ndiaye, Momo Banda Ndiaye, Alle Gueye, Serigne Saliou Mbacke, Fatou Kane, Mohamed Inejih El Ghouriechy, Lala Bouna Seck, Ndiaga Matar Gaye, Amadou Alpha Sall, Moustapha Ndiaye, Ousmane Faye, Amadou Gallo Diop, Jean-Michel Heraud

**Affiliations:** 1Clinique de Neurosciences Ibrahima Pierre Ndiaye, Centre Hospitalier National Universitaire de Fann, Dakar 10700, Senegal; jamilkahwagi@gmail.com (J.K.); mbodjahmadoubamba@yahoo.fr (A.B.M.); dabaya16.rd@gmail.com (R.D.); bayass2@hotmail.fr (E.h.M.); momobndiaye@gmail.com (M.B.N.); allegueye5@hotmail.com (A.G.); mbackesaliou236@gmail.com (S.S.M.); fkane29@gmail.com (F.K.); medinejih111@gmail.com (M.I.E.G.); lalasec@yahoo.fr (L.B.S.); ndiagamatar@gmail.com (N.M.G.); agallodiop@gmail.com (A.G.D.); 2Virology Department, Institut Pasteur de Dakar, 36, Avenue Pasteur, BP 220, Dakar 12900, Senegal; ousseynou.seye.2310@gmail.com (A.O.S.); martin.faye@pasteur.sn (M.F.); gamou.fall@pasteur.sn (G.F.); ndongo.dia@pasteur.sn (N.D.); babacar.ndiaye@pasteur.sn (B.N.); amadou.sall@pasteur.sn (A.A.S.); ngouille@hotmail.com (M.N.); ousmane.faye@pasteur.sn (O.F.); 3Centre Hospitalier National de Pikine, Pikine 16000, Senegal; fall.maouly@gmail.com; 4Institut Pasteur de Madagascar, Antananarivo 101, Madagascar; soafy@pasteur.mg; 5Encephalitis Society, North Yorkshire YO17 7DT, UK; ava@encephalitis.info; 6Department of Clinical Infection, Microbiology and Immunology, University of Liverpool, Liverpool L69 7ZX, UK

**Keywords:** encephalitis, virus, herpetic viruses, SARS-CoV-2, Senegal, Africa

## Abstract

The burden of encephalitis and its associated viral etiology is poorly described in Africa. Moreover, neurological manifestations of COVID-19 are increasingly reported in many countries, but less so in Africa. Our prospective study aimed to characterize the main viral etiologies of patients hospitalized for encephalitis in two hospitals in Dakar. From January to December 2021, all adult patients that met the inclusion criteria for clinical infectious encephalitis were enrolled. Cerebrospinal fluids, blood, and nasopharyngeal swabs were taken and tested for 27 viruses. During the study period, 122 patients were enrolled. Viral etiology was confirmed or probable in 27 patients (22.1%), with SARS-CoV-2 (n = 8), HSV-1 (n = 7), HHV-7 (n = 5), and EBV (n = 4) being the most detected viruses. Age groups 40–49 was more likely to be positive for at least one virus with an odds ratio of 7.7. The mortality was high among infected patients, with 11 (41%) deaths notified during hospitalization. Interestingly, SARS-CoV-2 was the most prevalent virus in hospitalized patients presenting with encephalitis. Our results reveal the crucial need to establish a country-wide surveillance of encephalitis in Senegal to estimate the burden of this disease in our population and implement strategies to improve care and reduce mortality.

## 1. Introduction

Encephalitis is an inflammation of the cerebral parenchyma that usually presents with a combination of clinical signs such as fever, headache, loss of consciousness, seizures, personality change, focal neurological deficits, and coma associated most of the time with biological abnormalities (pleocytosis) and specific neuroimaging and electroencephalogram [1,2]. Forty to sixty percent of encephalitis cases are unexplained [3,4,5]. Encephalitis can be caused by pathogens (bacteria, fungi, viruses, parasites) or immune system dysregulation. It is estimated that the prevalence of infectious encephalitis and autoimmune encephalitis are comparable [6]. Regarding infectious encephalitis, a viral infection occurs in 20% to 50% of the cases [5], and with the improvement of immunization against meningitis in Africa, it is expected that viruses will probably be the main causes of infectious encephalitis. Herpes Simplex Virus type 1 (HSV-1) is the most frequently detected viral agent in cases of fatal encephalitis in the USA and Europe [1,3,7,8,9]. However, there are other family viruses associated to encephalitis (Herpesviridae, Flaviridae, Picornaviridae, Rhabdoviridae, Orthomyxoviridae, etc.) [1,10,11]. More recently, infections due to the pandemic virus SARS-CoV-2 responsible for COVID-19 disease have been associated with different neurological manifestations including encephalopathy, encephalitis and cerebrovascular pathologies, and acute myelitis [12,13,14,15]. A retrospective study showed significant neurological effects in patients with COVID-19 up to six months post-infection [16].

The identification of a specific agent remains important, and modern molecular diagnostic methods have made it possible to identify many infectious agents associated with encephalitis [17,18,19]. Unfortunately, some limitations, such as the cost of the analyses, their relative sensitivity and specificity, do not allow routine use, especially in low-resource countries. This most often leads clinicians who manage cases of encephalitis to adopt a probabilistic approach.

To date, there is very little data on the etiology of encephalitis in Senegal and more broadly in Sub-Saharan Africa, and the real burden of viral-associated encephalitis is almost unknown. A review of pathogens detected in patients with encephalitis and meningitis in Western Africa showed that most of the studies focused on meningitis, and thus reported mostly bacterial infection [20]. Less than 0.1% of total cases from 139 published studies were identified as viral encephalitis. In Uganda, some cases of encephalitis associated with cytomegalovirus (CMV), enterovirus, HSV, Human Herpes Virus type 6 or 7 (HHV-6/7), and Varicella Zoster virus (VZV) have been reported [17]. In Senegal, a study conducted in 30 children with meningitis hospitalized at Diamniadio Children’s Hospital, was able to detect nine enteroviruses and one CMV from cerebrospinal fluid (CSF) [21]. Moreover, a neonatal Parechovirus meningitis was described in 2018 in a ten-day-old infant [22]. To fill the gaps regarding viruses associated with encephalitis in Senegal, we conducted a study aimed at identifying the main viral etiologies associated with encephalitis and describing the clinical presentation of patients.

## 2. Methods

### 2.1. Study Design

From January to December 2021, we conducted a prospective study in two neurology wards of two hospitals located in Dakar (CHNU of Fann) and in Pikine (National Hospital of Pikine). Patients presenting clinical signs of encephalitis assessed by clinicians and requiring hospitalization were sampled and tested, as part of routine management of encephalitic patients. Patients presenting with an identified cause (e.g., Malaria) were excluded from the study. Sampling consisted of cerebrospinal fluids (CSF) obtained from a lumbar puncture (if no other contraindications), blood specimens and nasopharyngeal (NP) swabs placed in viral transport media. Specimens were shipped the same day to the medical laboratory and the national reference laboratory for rabies and viral encephalitis located at the Institut Pasteur de Dakar. Some patients also benefited from specific neuroimaging (brain, medullary and thoracic) as well as electroencephalogram (EEG).

### 2.2. Biological Testing

Cerebrospinal fluids (CSF) were analyzed in cytology, biochemistry, bacteriology, parasitology, and mycology. Leftover CSFs were tested for the presence of viruses using real-time (RT)-PCR. Briefly, viral DNA/RNA were extracted from 200 µL of CSF and NP using Veri-Q PREP M16 nucleic acid extraction system (MiCoBioMed, Gyeonggi-do, South Korea) according to manufacturer’s instructions. Viral RNA was extracted from blood specimens (sera) using the QIAmp Viral RNA Mini Kit (QIAGEN, Hilden, Germany) according to the manufacturer’s instructions. RNA was eluted in 60 μL of nuclease-free water and stored at −80 °C until testing.

Nucleic acids were tested for a panel of 27 viruses using different methods with different sensitivities as seen in Appendix A. Combined nasopharyngeal and oropharyngeal swabs were collected and tested for SARS-CoV-2 as well as a panel of Respiratory viruses. Real-time PCR assays were conducted according to manufacturer’s instruction for commercial kit or as previously published [23,24,25,26,27,28,29], and already used in routine diseases surveillance reference laboratories at the Institut Pasteur de Dakar. For viruses included into the Allplex Meningitis-V1 and -V2 IVD kits (Seegene Inc., Seoul, Republic of Korea), PCR experiments were performed using the CFX96 Dx Real-Time PCR Detection Systems for in vitro Diagnostics (IVD) (Bio-Rad, Singapore). PCR runs were then analysed using the IVD Seegene Viewer for Real time Instruments program ver 3.24.000 (Seegene Inc., Seoul, Republic of Korea). For all other viruses, PCR experiments were performed using either 7500 or QuantStudio 5 Real Time PCR System (Applied-Biosystems, Thermo Fisher Scientific, Waltham, MA, USA).

### 2.3. Interpretation of Virological Results

A viral encephalitis was defined as **confirmed** when a patient presented with clinical signs of encephalitis, and this was confirmed by viral detection in CSF.A viral encephalitis was defined as **probable** when a patient presented with clinical signs of encephalitis and this was confirmed by viral detection in specimens outside the CSF (i.e., NP swabs or blood).

### 2.4. Neurological Assessment of Patient at Discharge

Surviving patients that were positive for at least one virus, were assessed for neurological status before discharge. Study doctors examined patients and neurological sequelae were assessed using the Liverpool Outcome Score (LOS) for assessing adults at Discharge [30]. The LOS assesses, through a series of 10 questions, basic motor, cognitive and behavioral functions posed to patient and/or caregiver. The clinicians then give a score regarding abilities of patient for simple activities (sitting, standing up, walking, hands on head and picking up). The final outcome score for each subject was the lowest score received for any question. Scores range from five (full recovery) to two (severe sequelae likely to make the patient dependent). The score of one is associated to death. The form used by clinicians in our study is available at the following address: https://www.liverpool.ac.uk/media/livacuk/infectionandglobalhealth/braininfections/DischargeLOS_ADULTS.pdf (accessed on 6 March 2022).

### 2.5. Statistical Analyses

Categorical variables were reported as counts and percentages. Comparisons were conducted using a chi-square or Fisher’s exact test, as appropriate. Comparisons of continuous variables were conducted using a Wilcoxon test. Two-sides *p* values < 0.05 were considered statistically significant. The degree of association between positivity for viral infection was expressed as an odds ratio (OR) considering potential confounding factors. All statistical analyses were performed using R software version 4.0.3 (Vienna, Austria) [31].

## 3. Results

### 3.1. Characteristics of Patients and Specimens

From January to December 2021, a total of 122 patients were enrolled and tested. The sex ratio male to female was 57/65 (0.88) (Table 1). The mean age of patients was 41.8 years (95% CI: 39.0–44.6), and 68.9% (84/122) had fever. Although mean age of women (42.7 years) was greater than men (40.7 years), no statistical difference was found (*p*-value = 0.45). When looking at biological parameters of CSF, 62.8% (71/113) had less than five elements per µL, and 75.2% (85/113) had less than 50% of lymphocytes. Biochemistry analysis of CSF revealed that 71.9% (82/114) of patients had normal glycorrhachia and 73.7% (84/114) had normal CSF protein level (Nine and eight data were missing for biological and biochemistry parameters, respectively). No difference was observed for fever and all biological parameters of CSF according to the sex of patients.

### 3.2. Characteristics of Infected Patients

Overall, 27 (22%) patients tested positive for at least one virus (Table 2). Although positivity rate was higher in women (25%), no statistical difference was noted when compared to men (19%). Age group 40–49 was more likely to be positive for at least one virus with adjusted OR of 7.7 [95% CI: 1.8–41.7]. No statistical difference was observed according to age and gender.

When looking at other parameters, multivariate analysis showed that patient with more than 99 leukocytes per µL in their CSF, were more likely to be infected with a virus with adjusted OR of 25.4 [95% CI: 5.5–153.4] (Table 2).

Amongst the 27 infected patients, we detected a total 30 pathogens of which 28 were viruses, one fungus (*Aspergillus* spp.) and one parasite (*Cryptococcus* spp.) (Table 3). Three patients were co-infected. (VZV and SARS-CoV-2, Epstein-Barr virus EBV and *Aspergillus* spp., and HSV-1 and *Cryptococcus* spp.). We confirmed a viral etiology in 18 (14.8%) patients, while the other nine patients (7.4%) had a probable viral encephalitis. SARS-CoV-2 (n = 8) was the main virus detected followed by HSV-1 (n = 7), HHV-7 (n = 5), EBV (n = 4), VZV (n = 2), and Rhinoviruses (n = 2). The majority of patients with a viral etiology had fever (77.8%, 21/27) and a pleocytosis in CSF (64.0%, 16/25) (Two missing data).

### 3.3. Clinical Manifestations in Patients with a Viral Infection

Among hospitalized patients with a viral infection detected, the main neurological symptoms were impaired consciousness (52%), followed by motor deficit (37%), meningeal syndrome (37%), behavioral changes (30%), cranial nerve damage (30%), and seizures (26%) (Appendix A). The main extra-neurological symptoms included respiratory disorders (19%), arthralgia/myalgia (19%), and anosmia (11%). Most of the patients (70%) had no known underlying condition or comorbidities. Five patients (19%) had high blood pressure and three (11%) were immunocompromised (HIV-1). The fatality rate among positive patients was high and occurred rapidly since 11 of them (41%) died during hospitalization within four to seven days. The Liverpool Outcome Score (LOS) for assessing adults at discharge (ranging from four to 32 days) revealed that only four patients (15%) fully recovered after the infection (LOS = 5), eight (29%) had minor to moderate sequelae (LOS = 3 to 4), and four (15%) had severe sequelae at discharge (LOS = 2). Among the three immunocompromised patients, one was infected with VZV, one with EBV and another one was infected with both HSV-1 and *Cryptococcus* spp. Two of them died following the infection. For these three patients, the HIV-1 infection was first detected during the hospitalization, and patients were not under treatment against HIV. Unfortunately, we were not able to collect more data (CD4/CD8 count, course and stage of HIV-1 infection).

## 4. Discussion

Infectious encephalitis caused by viruses remains a public health problem worldwide but the real burden of encephalitis as well as its associated viral etiology remains poorly described in low- and middle-income countries. To our knowledge, our observational prospective study is one of the first that aimed to identify the main viral causes of infectious encephalitis in Senegal. During the study period, from January to December 2021, we enrolled 122 adult patients presenting symptoms of infectious encephalitis that required hospitalization. The gender difference among our patients reflected the sex ratio of the Senegalese population aged 20 years and more (sex ratio M/F = 0.90) [32]. Although no difference was observed according to sex, it is noteworthy that 87.5% (7/8) of positive patients aged 50 years and older were women. This could suggests that in Senegal, women older than 50 years are at a higher risk of developing a viral encephalitis. This observation is not described elsewhere and should be taken with caution. One explanation of the observed sex-disparity in our study could suggest the role of intrinsic genetic factors. It is known that immune response is different according to sex, which has an impact in susceptibility of women to some autoimmune diseases and viral infections [33]. Sex steroids decline more rapidly in aged women, leading to a decline in the immune system, and a greater occurrence of the chronic pro-inflammatory state is observed in women [34,35]. A pro-inflammatory state in older women associated with a viral infection that potentially produces a cytokine-storm like for SARS-CoV-2 could explain why 62.5% (5/8) of SARS-CoV-2 patients presenting an encephalitis in our study were women aged 50 years and more. Among positive patients, we noted that the age group 40–49 was more likely to be infected with at least one virus positive (OR = 7.7). In China, a nation-wide study on acute encephalitis also revealed that HSV infections were most frequent in adults aged from 18 to 59 years [36].

In terms of etiology, we detected a viral infection in 27 patients (22.1%), of which 17 had a virus detected in their CSF, nine had a virus detected in the respiratory tract and one had a coinfection with one virus detected in the CSF (VZV) and a virus detected in the respiratory tract (SARS-CoV-2). The case of coinfection including SARS-CoV-2 and VZV is interesting, as it may suggest that COVID-19 infection could play a role in the reactivation of some viruses of the *Herpesviridae* family. More studies are needed to address this question. The prevalence of viral infection observed amongst our encephalitic patients was in the range of previous studies conducted in USA [3,5], France [4], Vietnam [37,38], China [36] and Malawi [39]. Differences between studies are explained by the geographic region, the study population (children vs. adults) and the diagnostic tests used (PCR, serology, culture). Therefore, it is difficult to compare one study to another. In a retrospective study conducted from 2001 to 2003 at the Fann Hospital in Dakar, Soumaré et al. have mainly reported parasites (*Plasmodium* spp. and *Cryptococcus* spp.) and bacteria (*Pneumococcus* spp. and *Meningococcus* spp.) as the main infectious pathogens in 470 patients hospitalized for neuromeningeal disease [40]. They have only identified four cases (0.9%) of viral encephalitis based on clinical manifestation. Our study focused on viral infection, and for that reason, we excluded patients with a rapid test positive for malaria. We also used molecular assays that are more sensitive, explaining why we were able to identify a higher number of viral etiologies among our patients.

We conducted our prospective study during two waves of SARS-CoV-2 that affected Senegal from January to March and from July to August 2021. Since it is now known that SARS-CoV-2 can affect the central nervous system (CNS), leading to severe neurological diseases including encephalitis [12], we tested all our patients to see whether they were infected with this virus. Surprisingly, out of our 122 patients enrolled, eight tested positive for COVID-19. Although no SARS-CoV-2 genome was detected in the CSF, but only from nasopharyngeal swabs, we considered that these SARS-CoV-2 associated encephalitis were highly probable. Several studies have reported neurologic disease following SARS-CoV-2 infection [12,13,14,15]. We confirmed this observation in Dakar (Senegal), and we consider that the real burden of neurologic disease associated with SARS-CoV-2 infection should be investigated in our region and more broadly in Africa. An interesting case was a co-infection of a patient with SARS-CoV-2 (in NP) and VZV (in CSF). These results could suggest that SARS-CoV-2 infection could reactivate herpetic viruses like VZV, probably due to the neuro-inflammation and immune system impairment caused by SARS-CoV-2. Although our cohort was small, since most of our patients (6/8; 75%) with a SARS-CoV-2 associated encephalitis were female, we should investigate if females are more at risk of developing neurologic syndromes following SARS-CoV-2 infection.

In countries where molecular tests are not available, it is difficult for clinicians to detect viral associated encephalitis. We tried to identify some good predictors to assist clinicians with decision-making. We observed that patients presenting with pleocytosis greater than 99 elements per microliter were more likely to be infected with a virus (OR = 25.4 (*p* < 0.001)). A previous study conducted in Vietnam showed that viral diagnostic yield increased when results from CSF including pleocytosis are used for the case definition [37]. Nevertheless, as mentioned by these authors and as observed in our study, a still significant proportion of patients did not meet the case definition (e.g., no fever, no pleocytosis, normal CSF parameters). This highlights the need to conduct more work and develop new tools to optimize the case definition and prognosis of viral encephalitis.

We observed a high mortality rate among positive patients, with 11 (41%) deaths during hospitalization. This mortality rate is explained by several factors. First, the delay between first signs and hospitalization. Indeed, most of the included patients arrived at the hospital after more than five days following onset of disease. Second, some specific treatment like intravenous acyclovir is not available in Senegal to treat patients with HSV or VZV infection, and clinicians can only use oral acyclovir or valacyclovir. Finally, even if a treatment is made available to patients, the price of this antiviral is too expensive for most of our patients. Although intravenous acyclovir remains the gold standard to treat HSV and VZV infection and the use of oral valacyclovir should be considered with caution [41], some authors have suggested that oral valacyclovir could be an alternative treatment for patients in low- and middle-income countries [42]. Besides this mortality rate, the first use of the Liverpool Outcome Score for adults at discharge in our setting revealed that less than 40% of infected patients with encephalitis fully recovered or had minor sequelae at discharged. A significant number of patients (22%) had moderate to severe sequelae, which will probably impact the patient and their dependents socio-economically. Unfortunately, we were unable to complete a further LOS assessment during a follow up of patients, and therefore, the duration of sequelae observed at discharged could not be addressed, but is recommended for future studies.

Our study has some limitations. Due to socio-economic factors of patients, we could not use imaging (e.g., Scanner) as part of our case definition. Thus, the enrollment of patients was mainly based on suspicion from clinicians according to clinical presentation (e.g., altered mental status for more than 24 h with or without fever). Consequently, our prevalence rate of viral encephalitis might be underestimated, as we may have included patients that would have been excluded in other settings according to the results of imaging. Another limitation is the general low viral load in CSF, transient viral excretion, and delays between onset of disease and hospitalization. Although real-time PCR is highly sensitive and recommended for viral encephalitis [18], we cannot exclude some false negatives, and consequently we may have underestimated the real viral prevalence. Nevertheless, our prevalence is in the same range compared to other studies conducted in different settings, thus we are confident with our results. Finally, we have tested patients for a panel of viruses selected because of their occurrence in viral encephalitis demonstrated in previous studies (e.g., herpetic viruses). We also tested for arboviruses because they circulate in Senegal and have been involved in viral encephalitis (e.g., West-Nile virus and Zika). However, many other viruses could also be tested by PCR, as well as bacteria, parasites, and fungus. We are planning to explore in more detail all negative patients with high suspicion of viral encephalitis using a next-generation sequencing approach.

## 5. Conclusions

Viral encephalitis is a public health problem in Senegal and probably in Africa. With a viral positivity rate of 22% and a mortality rate around 41% among patients suspected of infectious encephalitis, our study shows the urgent need for health authorities to invest in the surveillance of encephalitis and the management of patients. Intravenous antivirals should be made available in the country at low cost to reduce the mortality and sequelae associated with viral encephalitis. To date, no data are available regarding number of death that could be associated to encephalitis and in particular viral encephalitis. Thus, more analyses and work are required to estimate the real burden of viral encephalitis in Senegal.

## Figures and Tables

**Table 1 viruses-14-00871-t001:** Characteristics of encephalitis patients enrolled in the study according to sex, Dakar, January–December 2021.

Characteristic	Male N (%)	Female N (%)	Total N (%)	*p*-Value ^2^
**Patient**	57 (46.7)	65 (53.3)	122 (100)	
**Age**				
Mean (95% CI)	40.7 (36.8–44.7)	42.7 (38.8–46.7)	41.8 (39.0–44.8)	0.45
**Fever**				
**Yes**	39 (46)	45 (54)	84 (100)	
**No**	18 (47)	20 (53)	38 (100)	
**Pleocytosis (nb/µL)**				
[0–4]	32 (45)	38 (55)	71 (100)	
[5–99]	16 (53)	14 (47)	30 (100)	
>99	6 (50)	6 (50)	12 (100)	
missing	3 (33)	7 (67)	9 (100)	
**% Lymphocytes ^1^**				
[0–50]	41 (48)	44 (52)	85 (100)	
[50–100]	13 (46)	15 (54)	28 (100)	
missing	3 (33)	6 (67)	9 (100)	
**Glycorrhachia (g/L)**				
[0–0.4]	7 (47)	8 (53)	15 (100)	
[0.4–0.8]	40 (49)	42 (51)	82 (100)	
>0.8	8 (47)	9 (53)	17 (100)	
missing	2 (25)	6 (75)	8 (100)	
**Proteinorrachia (g/L)**				
[0–1.0]	38 (45)	46 (55)	84 (100)	
>1.0	17 (57)	13 (43)	30 (100)	
missing	2 (25)	6 (75)	8 (100)	

^1^ The proportion (%) of lymphocytes in CSF were calculated only when pleocytosis was equal or greater than five elements/µL. ^2^ Only *p*-value < 0.05 considered as statistically significant are shown.

**Table 2 viruses-14-00871-t002:** Characteristic of encephalitic patients according to viral detection, Dakar, January–December 2021.

Characteristic	Positive N (%)	Negative N (%)	Total N (%)	*p*-Value ^2^	Adjusted OR (95% CI)
**Total**	27 (22)	95 (78)	122 (100)		
**Sex**					
Male	11 (19)	46 (81)	57 (100)		
Female	16 (25)	49 (75)	65 (100)		
**Age (Year)**					
20–29	4 (12)	29 (88)	33 (100)		
30–39	5 (18)	23 (82)	28 (100)		
40–49	10 (45)	12 (55)	22 (100)	0.01	7.7 [1.8–41.7]
50–59	5 (28)	13 (72)	18 (100)		
>59	3 (14)	18 (86)	21 (100)		
Mean [95% CI]	44.0 [38.5–49.4]	41.3 [38.0–44.5]	41.8 [39.0–44.6]		
**Fever**					
Yes	21 (25)	63 (75)	84 (100)		
No	6 (16)	32 (84)	38 (100)		
**Pleocytosis (nb/µL)**					
[0–4]	9 (13)	62 (87)	71 (100)		
[5–99]	7 (23)	23 (77)	30 (100)		
>99	9 (75)	3 (25)	12 (100)	<0.001	25.4 [5.5–153.4]
missing	2 (22)	7 (78)	9 (100)		
**% Lymphocytes ^1^**					
[0–50]	12 (14)	73 (86)	85 (100)		
[50–100]	13 (46)	15 (54)	28 (100)		
Missing	2 (22)	7 (78)	9 (100)		
**Glycorrhachia (g/L)**					
[0–0.4]	6 (40)	9 (60)	15 (100)		
[0.4–0.8]	16 (20)	66 (80)	82 (100)		
>0.8	3 (18)	14 (82)	17 (100)		
missing	2 (25)	6 (75)	8 (100)		
**Proteinorrachia (g/L)**					
[0–1.0]	14 (17)	70 (83)	84 (100)		
>1.0	11 (37)	19 (63)	30 (100)		
missing	2 (25)	6 (75)	8 (100)		

^1^ The proportion (%) of lymphocytes in CSF were calculated only when pleocytosis was equal or greater to five elements/µL. ^2^ Only *p*-value < 0.05 considered as statistically significant are shown.

**Table 3 viruses-14-00871-t003:** Pathogens detected in encephalitic patients according to sex, Dakar, January–December 2021.

Pathogens	Male N (%)	Female N (%)	Total N (%)	Etiology
**Viruses** ^1^				
SARS-CoV-2	2 (25)	6 (75)	8 (100)	Probable
HSV-1	5 (71)	2 (29)	7 (100)	Confirmed
HHV-7	3 (60)	2 (40)	5 (100)	Confirmed
EBV	1 (25)	3 (75)	4 (100)	Confirmed
VZV	1 (50)	1 (50)	2 (100)	Confirmed
Rhinovirus	0 (0)	2 (100)	2 (100)	Probable
**Fungus**				
*Aspergillus* spp.	0 (0)	1 (100)	1 (100)	Confirmed
**Parasites**				
*Cryptococcus* spp.	1 (100)	0 (0)	1 (100)	Confirmed
**Total** ^2^	**13 (43)**	**17 (57)**	**30 (100)**	

^1^ HSV = Herpes Simplex Virus; EBV = Epstein-Barr Virus; HHV = Human Herpes Virus; VZV = Varicella-Zosters Virus. ^2^ The total number of pathogens detected is greater than the total number of positive individuals because of three coinfections.

## Data Availability

Not applicable.

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
