# Peer review of "Surveillance of Viral Encephalitis in the Context of COVID-19: A One-Year Observational Study among Hospitalized Patients in Dakar, Senegal"

_viruses, 2022, doi:10.3390/v14050871_

Round 1

Reviewer 1 Report

In this manuscript, Kahwagi et al. present a statistic report of encephalitis cases and the potential viral etiology based on patients from two hospitals in Dakar, Senegal from January to August of 2021. Through this small cohort study (74 patients in total), the authors report a substantial portion of patients (23 out of 74) with viral infections, based on detection of viral nucleic acids. The potential viral etiology includes SARS-CoV-2 and several different herpesviruses. Interestingly, SARS-CoV-2 is the most frequently detected in patients with encephalitis, suggesting a correlation between neurological manifestation and COVID-19 infection. This is in line with the growing reports of neurological symptoms in COVID-19 patients. Besides, the coinfection of SARS-CoV-2 with herpesviruses also suggests the manipulation of host immunity by SARS-CoV-2 that enables the reactivation of these commonly found viruses in latency. Moreover, the data also reveals people in the age of 40-49 and 50-59 are more likely to be infected by at least one of these viruses, suggesting a correlation with declining immunity in higher-ages. Overall, this work is quite simple and clear, and the proposed “conclusions” are of potential interest for guiding the setup of public health system in related region/country. The major limitation is the small sample size, making the analyses not sufficiently evident to derive rather confident conclusions. By and large, this study is refrained to descriptive study and indicative analysis, but failed to permit more in depth insights underlying these observations.

Reviewer 2 Report

Despite numerous studies, the knowledge of the etiology of viral encephalitis leaves much to be desired.   Therefore, the presented study on the African population is original and potentially interesting for scientists and physiciants. Before publishing it would be recommendable to introduce some changes: 1. The introduction and discussion should be significantly shortened, 2. Conclusions should refere strictly to results presented, 3. Methods section: it is unclear whether CSF from all subjects was tested (line 110), "some" - how many (line 113), number of subjects 70 or 74 (lines 167 vs. 168), proteinorachia (line 174) better CSF protein level, sensitivity of employed molecular motods should be also presented.
The comment on HIV-positive patients is necessary.  

Author Response

We are grateful to the Editor and reviewers for their encouraging comments. We have revised substantially our manuscript to address all comments from reviewers. We have increased the sample size of patients resulting in some changes regarding some results as explained below. We think that our revised manuscript has greatly improved compared to the first version and we hope that it will be considered as suitable for publication.

Reviewer 2:

Despite numerous studies, the knowledge of the etiology of viral encephalitis leaves much to be desired. Therefore, the presented study on the African population is original and potentially interesting for scientists and physicians. Before publishing it would be recommendable to introduce some changes:

1. The introduction and discussion should be significantly shortened,

Authors’ responses: We thank the reviewer for these suggestions. The introduction and conclusion have been shortened substantially. It is to be noted that the entire manuscript has been checked by two different English native speakers.

2. Conclusions should refer strictly to results presented,

Authors’ responses: We understand the point of view of the reviewer, and we have made some revisions to delete part of the conclusion that did not refer to results presented. Since it is the first study of this type conducted in Senegal, we wanted to highlight in our conclusion important messages towards health authorities: (i) The importance to implement a surveillance of encephalitis to better estimate the prevalence of viral encephalitis and associated death; (ii) the need to make antivirals available in the country to reduce mortality and (iii) the implementation of a program to estimate epidemiologic and economic costs of infectious encephalitis in the country.

3. Methods section: it is unclear whether CSF from all subjects was tested (line 110), "some" - how many (line 113), number of subjects 70 or 74 (lines 167 vs. 168), proteinorachia (line 174) better CSF protein level, sensitivity of employed molecular methods should be also presented.

Authors’ responses:
- We are sorry for the confusion, and we will try to explain better to the reviewer. Indeed, all patients enrolled were supposed to have their CSF tested for Cytology (Leukocytes, parasites, bacteria, fungus), Biochemistry (glycorachia and proteinorachia), microbiology (bacterial, fungus culture) and virology. Although all virological tests were performed free charge of the patients, patients had to pay for some CSF analysis (cytology, biochemistry, and microbiology) either to the hospital or to themedical laboratory. For some patients it was not possible to pay for all these tests. This is the reason why, some data for CSF were missing and the total number CSF tested for virology is smoothly different than the total number of CSF tested for cytology and biochemistry. This information was noted as “Missing” in tables.
- The sensitivity of each molecular method was added in the supplementary Table 1. It is to be noted, that for some methods the diagnostic sensitivities were not assessed but the analytical limits of detection were available. Only for CCHFV, the method published and routinely used by the national reference laboratory did not mention the sensitivity and the limit of detection. We think that for the purpose of our study and considering that no CCHFV outbreaks was detected during the study period, it did not affect our results. - Proteinorachia in line 174 was replaced with CSF protein level

4. The comment on HIV-positive patients is necessary.

Authors’ responses: Since we included only three immunocompromised patients (HIV-1), we thought that it was not relevant to discuss this aspect. Following the suggestion of the reviewer, we added a comment on these HIV positive patients” (line 225-229)

Yours sincerely,
Dr Jean-Michel HERAUD, on behalf of all authors,

Round 2

Reviewer 1 Report

In the revised manuscript, the authors incorporated more subjects to increase the sample size of this analysis. Because of the new data, the derived conclusions are also updated accordingly. I appreciate the efforts made by the authors to improve the study even though still limited by the sample size of available data so far. Overall, I am contented with the revision.

Author Response

We thank the editor and reviewers for accepting our manuscript with minor revision.

We will address specifically the last comments from editor and reviewers#2:

Reviewer's comment:

Methods section: it is unclear whether CSF from all subjects was tested (line 110), "some" - how many (line 113), number of subjects 70 or 74 (lines 167 vs. 168), proteinorachia (line 174) better CSF protein level, the sensitivity of employed molecular methods should be also presented.

Authors’ responses:

  • We are sorry for the confusion. Indeed, according to our protocol, all patients enrolled were supposed to have their CSF tested for Cytology (Leukocytes, parasites, bacteria, fungus), Biochemistry (glucose, protein level), microbiology (bacterial, fungus culture), and virology. Although all virological tests were performed free of charge to the patients, patients had to pay for some CSF analysis (cytology, biochemistry, and microbiology) either at the hospital or at the medical laboratory. For some patients, it was not possible to pay for all these tests. This is the reason why some data for CSF were missing and the total number of CSF tested for virology is smoothly different than the total number of CSF tested for cytology and biochemistry. Amongst the 122 patients enrolled, 08 patients were not tested for biochemistry and cytology and one patient was not tested only for cytology. These missing data were noted as “Missing” in tables. We consider that these missing data did not affect the global analyses and our conclusion 
  • The sensitivity of each molecular method was added in the supplementary Table 1. It is to be noted, that for some methods the diagnostic sensitivities were not assessed but the analytical limits of detection were available. Only for CCHFV, the method published and routinely used by the national reference laboratory did not mention the sensitivity and the limit of detection. We think that for the purpose of our study and considering that no CCHFV outbreak was detected during the study period, it did not affect our results.

- Proteinorachia in line 174 was replaced with CSF protein level